# Determination of Neonicotinoid Insecticides in Environmental Water by the Enrichment of MIL-53 Mixed Matrix Membrane Coupled with High Performance Liquid Chromatography

**DOI:** 10.3390/ijerph20010715

**Published:** 2022-12-30

**Authors:** Gege Wu, Jiping Ma, Chenxi Wei, Shuang Li, Jinhua Li, Xiaoyan Wang, Lingxin Chen

**Affiliations:** 1School of Environmental & Municipal Engineering, Qingdao University of Technology, Qingdao 266033, China; 2CAS Key Laboratory of Coastal Environmental Processes and Ecological Remediation, Shandong Key Laboratory of Coastal Environmental Processes, Yantai Institute of Coastal Zone Research, Chinese Academy of Sciences, Yantai 264003, China; 3School of Pharmacy, Binzhou Medical University, Yantai 264003, China

**Keywords:** metal-organic framework, neonicotinoid insecticides, dispersive membrane extraction, high performance liquid chromatography, environmental analysis

## Abstract

Metal organic framework based mixed matrix membranes (MOF-MMMs) were synthesized and applied for dispersive membrane extraction (DME) of four neonicotinoid insecticides (nitenpyram, thiacloprid, imidacloprid, and acetamiprid) in environmental water, combined with high performance liquid chromatography (HPLC) for determination. Several experimental conditions were optimized in detail, involving dosage percentage of MOF, extraction time, sample pH, salinity, type and volume of eluent, and elution time. High sensitivity with limits of detection and quantification were achieved as 0.013–0.064 μg L^−1^ and 0.038–0.190 μg L^−1^, respectively, and good precision with relative standard deviations were obtained as 3.07–12.78%. The proposed method has been successfully applied to determine four neonicotinoid insecticides in tap water, surface water, and seawater, satisfactory recoveries of spiked water samples were between 72.50 and 117.98%. Additionally, the MOF-MMMs showed good reusability with the extraction efficiencies almost remaining stable after 14 cycles. The MOF-MMMs based DME followed by the HPLC method can be a promising utility for the determination of neonicotinoid insecticides in environmental water samples, with high sensitivity and convenient operation.

## 1. Introduction

Neonicotinoid insecticides, as a newest major class of synthetic organic insecticides, have undergone rapid development in recent years [1,2]. Low doses of neonicotinoid insecticides can be absorbed by plants and act quickly on insect pests and afford long-term protection for plants [3]. Due to their selectively high toxicity to insects, neonicotinoid insecticides have become the most widely used insecticides and account for about a quarter of all insecticides. Neonicotinoid residues can leach into water bodies by rainfall, and they have been frequently detected in groundwater, surface water, and ocean tides [4,5]. Neonicotinoid insecticides molecules contain ionizable basic amine or imine substituent, which can attack the acetylcholine receptors in the mammalian central nervous system, leading to adverse impacts on memory, cognition and behavior development of human beings [6]. Neonicotinoid contaminants in environments would pose a potential risk to both human health [7,8] and ecosystems [9]. Therefore, it is necessary to establish sensitive and simple analytical methods for the determination of trace neonicotinoid insecticides in environmental waters.

The residual concentration of neonicotinoid insecticides in environmental waters is generally low for commonly used instruments to detect. In general, the most commonly used analytical methods for neonicotinoid insecticides are high-performance liquid chromatography coupled with an ultraviolet detector (HPLC-UV) [10,11] or mass spectrometry (HPLC-MS) [12]. UV detectors are not competitive to provide high sensitivity as HPLC-MS. However, the cost of HPLC-MS analysis is high. HPLC-UV is more commonly available, and just needs a higher enrichment factor prior to analysis. Therefore, the sample pretreatment process is imperative to enrich target analytes and eliminate the matrix effects of complex samples. Several pretreatment methods, including packed solid-phase extraction (SPE) [13], dispersive SPE [14], magnetic SPE [15], and liquid-liquid microextraction [16] have been reported for the enrichment of neonicotinoid insecticides. In different SPE modes, the selection of adsorbent material is always the crucial factor to improve enrichment efficiency.

Metal-organic frameworks (MOFs) are a novel class of porous materials composed of organic ligands with inorganic clusters. Due to their unique properties, such as high specific surface area, tunable pore size, and easy functionalization [17,18], MOFs have attracted extensive attention and have been widely applied in various fields such as gas storage [19,20], catalysis [21,22], sensing [23], and especially in adsorbents [24,25,26]. Porous structure and abundant adsorption sites in the organic ligand and inorganic clusters make MOFs exhibit excellent adsorption performance towards various compounds. The separation of dispersive MOFs crystals from aqueous solutions is usually labored. Therefore, MOFs adsorbents are always combined with different SPE modes, such as SPE cartridge [27], solid-phase microextraction (SPME) [28], and stir bar sorptive extraction (SBSE) [29]. Our group has prepared magnetic MIL-101 and magnetic MOF-5 for magnetic SPE of pyrazole/pyrrole pesticides [30] and heterocyclic pesticides [31], respectively. However, amounts of MOFs remain in the sample solution and cause subsequent problems. The formation of the membrane is an efficient approach to integrating MOFs crystals, and meanwhile, the membrane makes the separation of MOFs from aqueous phase much more convenient [32,33,34].

In general, MOFs-based mixed-matrix membrane (MMM) is quite robust, in which, the MOFs particles act as a filler and the organic matrix acts as a continuous phase to ensure the continuity of the membrane. The MMM synthesis does not need substrates for MOF growth and it can obtain higher MOF loading [35]. The MOFs MMM have exhibited wide application prospects in gas separation and storage [36,37,38]. Meanwhile, the excellent adsorption capability and high porosity of MOFs MMM are also beneficial for targets’ mass transfer from the liquid phase to the MOFs MMM adsorbents [39]. In addition, the robust MOFs MMM makes the separation of adsorbents from aqueous solutions extremely convenient. Therefore, the MOFs MMM materials are potential candidate adsorbents in sample preparation. Among various series of MOFs, MILs are famed for their flexible skeleton and breathing effect. It has been proven that the flexible skeleton could significantly promote adsorption capacity [40]. In view of the electropositive basic amine or imine substituent on nitrogen heterocyclic ring in neonicotinoid insecticides, MILs MMM provides a good prospect for adsorption of neonicotinoid insecticides by cation-π bonding, π-π conjugation and breathing effect.

Therefore, herein, we proposed a kind of MOFs MMM (MIL-53 MMM) for simultaneous DME of four neonicotinoid insecticides followed by HPLC-DAD determination. The main influence parameters of the DME process were systematically investigated. Under optimal conditions, the developed DME-HPLC method was well validated and practically applied for the determination of neonicotinoid insecticides in different environmental waters.

## 2. Materials and Methods

### 2.1. Reagents and Materials

All chemicals were of at least analytical grade. Benzene-1,4-dicarboxylic acid (Tianjin Guangfu Fine Chemicals Research Institute, Tianjin, China), aluminum nitrate nonahydrate (Aladdin Reagent Co., Ltd., Shanghai, China) and N, N-dimethylformamide (Sinopharm Chemical Reagent Co., Ltd., Shanghai, China) were used to prepare MIL-53. Polyvinylidene Fluoride (PVDF) was purchased from American Arkema Company (Crosby, TX, USA). Methanol and acetonitrile were procured from Shanghai Anpu Reagent Co., Ltd. (Shanghai, China). Ethyl acetate was obtained from Honeywell (Charlotte, NC, USA). Acetone was supplied by Sinopharm Chemical Reagent Co., Ltd. Sodium hydroxide was purchased from Tianjin Hengxing Chemical Reagent Co., Ltd. (Tianjin, China). Sodium chloride was obtained from ShangHai Aibi Chemistry Preparation Co., Ltd. (Shanghai, China). Hydrochloric acid was purchased from Tianjin Kaixin Chemical Co., Ltd. (Tianjin, China). Ultrapure water (18.2 MΩ) was obtained by a model Millipore D-24 UV ultrapure water system (Millipore, France).

Imidacloprid, acetamiprid and thiacloprid were supplied by Shanghai Pesticide Testing Research Center (Shanghai, China). Nitenpyram was procured from Beijing Yinuokai Technology Co., Ltd. The structural formulas of four neonicotinoid insecticides are shown in Figure 1. Stock solutions at 1000 mg L^−1^ were prepared by dissolving nitenpyram, idacloprid, acetamiprid, and thiacloprid powders into methanol, respectively. Then, the stock solutions were stored at 4 °C.

The environmental water samples were collected from laboratory tap water, Qingdao Bohai Bay seawater and Qingdao Yinfu Reservoir surface water. All samples were filtered through the 0.45 μm filter membrane, and then placed in the refrigerator at 4 °C for further analysis.

### 2.2. Apparatus

The concentration of four neonicotinoid insecticides was detected by Agilent 1100 liquid chromatographic system coupled with DAD detector. HPLC separation was carried out using a ZORBAX SB-C18 column (4.6 × 250 mm, 5 µm) at room temperature. The sample injection volume was 20 µL. DAD absorbance was monitored at 244 nm and 270 nm. The mobile phase was a mixture of methanol and water. Gradient elution conditions were as follows: 0–5 min, isocratic 30% methanol; 10–18 min, isocratic 55% methanol. The flow rate was 0.8 mL min^−1^. Under these optimum conditions, all studied insecticides were well separated from each other.

SUPRA 55 scanning electron microscope (SEM, ZEISS, Germany) were used to characterize the synthesized MIL-53 mixed matrix membrane. X-ray diffractometer (XRD) pattern recorded on D8 Advance (Bruker, Billerica, MA, USA). Frontier Nicolet iN10 infrared spectrometer (Thermo Fisher, Waltham, MA, USA) with an attenuated total reflection (ATR) accessory was used to identify the functional groups of materials.

### 2.3. Synthesis of MIL-53-PVDF MMM

MIL-53 was synthesized by the reported solvothermal method [41] 3.376 g aluminum nitrate nonahydrate and 0.996 g terephthalic acid were dissolved in 44 mL DMF and 16 mL ultrapure water. The mixture was placed in a water bath and stirred at 40 °C for 2 h. Then the mixed solution was sealed in a 100 mL Teflon-lined stainless steel autoclave and heated in the oven at 130 °C for 48 h. After cooling naturally to room temperature, the obtained particles were collected by centrifugation and washed three times with DMF. The resulting MIL-53 was dried at 100 °C.

Subsequently, 120 mg of MIL-53 powder was dispersed in 5 mL of acetone under an ultrasonic bath for 30 min. Then, 2 mL of PVDF solution (50 mg PVDF dissolved in 2 mL DMF) was added to the MOFs suspension. The mixed suspension of MOFs and PVDF was further mixed under ultrasound for 30 min, and then rotary evaporation was used to remove acetone. The obtained mixture of MIL-53 and PVDF in DMF was uniformly poured onto a clean glass plate. The coated glass substrate was then heated at 70 °C, and the MIL-53 membrane gradually formed when the solvent was dried. The synthesis procedure is schematically shown in Figure 2.

### 2.4. DME Procedure

A 100 mL water sample containing a certain concentration of neonicotinoid insecticides was placed in a beaker and the MIL-53 MMM was immersed into the aqueous solution, the beaker was shaken for 35 min. After extraction, 2 × 5 mL acetone was used to elute the adsorbed insecticides from MIL-53 MMM, elution time was 6 min. The collected extract was concentrated by a gentle stream of nitrogen to approximately dry, then it was diluted with 0.5 mL ultrapure water/methanol (*v*/*v* = 7:3), which was filtered with 0.45 μm nylon membrane prior to HPLC analysis. The extraction procedure is shown schematically in Figure 3.

## 3. Results and Discussion

### 3.1. Choice of Membrane Material

MIL-53 materials are a classic series of MOFs with flexible frameworks when they interact with polar gas molecules by hydrogen bonds [41]. The relevant literature reported that MIL-53 exhibited higher affinity to compounds with higher polarity [42]. Taking account of the cationic amine or imine groups in neonicotinoid insecticides, MIL-53 was selected as the adsorbents, and the abundant benzene rings in MIL-53 were also beneficial to form π-π conjugation and cationic-π bonding to trap neonicotinoid insecticides. On the other hand, MIL-53 MMM materials were proven to be stable when applied in the desalination of dyes contaminated water [43]. Therefore, MIL-53 MMM based DME pretreatment technique provides an appropriate approach to enriching neonicotinoid insecticides from water samples.

### 3.2. Characterization of MIL-53 MMM

The morphology of MIL-53 MMM was characterized by SEM. As seen in Figure 4, cubic crystals of MIL-53 were cross-linked with each other by the polymer binder. The microscopic morphology of MIL-53 MMM are consistent with the reported literature [44], indicating successful synthetization.

FT-IR spectra of MIL-53 (a) and MIL-53 MMM (b) were shown in Figure 5. As can be seen from Figure 5a, the absorption peaks at 1500 cm^−1^ and 1450 cm^−1^ could be assigned to C=C stretching in the benzene ring, and the absorption band at 1696 cm^−1^ could be attributed to C=O groups in terephthalic acid. These results demonstrated that the terephthalic acid ligand was incorporated in the framework of MIL-53. Above absorption peaks were also exhibited in MIL-53 MMM (Figure 5b), suggesting that the chemical structure of MIL-53 remains in MIL-53 MMM. On the other hand, absorption band at 1200 cm^−1^ related to C-F stretching vibration in PVDF. These results indicate that MIL-53 was embedded in the PVDF membrane and MIL-53 MMM was successfully prepared [45].

XRD patterns of the MIL-53 crystal (a) and MIL-53 MMM (b) were shown in Figure 6. As seen, the observed diffraction peaks at 2θ = 9.3°, 17.8°, 21.8°, and 27.2° in the MIL-53 crystal were consistent with the reported literature [46]. These characteristic peaks also existed in MIL-53 MMM, indicating that the crystal structure of MIL-53 was remained in the process of forming a membrane.

### 3.3. Optimization of DME Conditions

Several major factors that could affect extraction efficiency were investigated, including dosage ratio of MOFs in MOF-MMM, extraction time, sample pH, salt concentration, type and volume of desorption solvent, and elution time. Further, 10 μg L^−1^ of spiked ultrapure water sample was used under different experimental conditions, and the optimization was conducted by three parallel experiments.

#### 3.3.1. Effect of the Dosage of MOFs 

The dose ratio of MOFs in MOF-MMM is one of the key parameter conditions that affect the efficiency of the DME process. Different amounts of MIL-53 powder in the range from 30 mg to 150 mg were mixed with 50 mg of PVDF to form the MMM. As shown in Figure 7a, the peak areas of the four neonicotinoid insecticides gradually increased from 30 mg to 120 mg, and then slightly decreased from 120 mg to 150 mg. This may be due to the weight of the whole MIL-53 MMM being much too heavy, and cannot be uniformly suspended in water samples, leading to a decrease in extraction efficiencies. As a result, 120 mg of MIL-53 powder was set to prepare the MIL-53 MMM.

#### 3.3.2. Effect of Extraction Time

Sufficient extraction time is significant to reach the adsorption equilibrium between sample solutions and adsorbents. Different extraction time at 15 min, 25 min, 35 min, 45 min, and 50 min were investigated. As shown in Figure 7b, the peak areas of the four neonicotinoid insecticides increased rapidly from 15 min to 35 min and then reached a plateau. Therefore, 35 min was selected as the extraction time.

#### 3.3.3. Effect of Sample Solution pH

The pH of the sample solution not only affects the molecular forms of neonicotinoid insecticides, but also affects the surface charge of the MIL-53 MMM. In this study, solution pH value was adjusted in the range of 3–9 by 0.1 M hydrochloric acid and sodium hydroxide. As shown in Figure 7c, the peak areas of four neonicotinoid insecticides remained almost constant within pH 3.0–8.0. The main adsorption mechanism in this pH range were π-π conjugation between benzene rings in MIL-53 and neonicotinoid insecticides, and cationic-π bonding between electropositive basic amine or imine in neonicotinoid insecticides and benzene rings in MIL-53. When pH was lower than 4.0, the neonicotinoid insecticides and MIL-53 MMM were both positively charged, the electrostatic repulsion leading to the decrease in extraction efficiencies. With the pH value increasing higher than 8.0, more anionic OH- species in the sample solution would connect with neonicotinoid insecticides, cationic-π bonding was reduced, leading to the decrease in extraction efficiencies. Therefore, the pH value of the water sample in the further experiments was not adjusted.

#### 3.3.4. Effect of Salt Concentration

The salt concentration can influence the ionic strength and viscosity of the sample solution, which would affect the interactions between the sorbent and analytes. In this study, different salt concentrations of 0%, 0.1%, 1%, 5%, and 10% were investigated by adding sodium chloride to the water samples. As shown in Figure 7d, the peak area of the four neonicotinoid insecticides slightly decreased with the increase in salinity. This phenomenon can be explained as the dissociated NaCl surrounding the MIL-53 MMM would impede the cationic-π bonding between neonicotinoid insecticides and benzene rings in MIL-53. Therefore, no salt was added to the water sample in subsequent studies.

#### 3.3.5. Effect of Desorption Condition

The property of desorption solvent is an important factor affecting elution efficiency. Four frequently used organic solvents, including methanol, acetonitrile, ethyl acetate, and acetone, were examined as desorption solvents. As shown in Figure 7e, acetone had the best elution efficiency for four neonicotinoid insecticides. Therefore, acetone was selected as the desorption solvent for further experiments.

Due to the fresh solvent can desorb more analytes, the eluent process was conducted twice. Thus, 2 × 2 mL, 2 × 3 mL, 2 × 4 mL, 2 × 5 mL, and 2 × 7 mL of acetone were used to optimize the volume of desorption solvent. As shown in Figure 7f, the peak areas of four neonicotinoid insecticides gradually increased with the volume of acetone increasing from 2 × 2 mL to 2 × 5 mL and then reached to a plateau. Indicating that 2 × 5 mL of acetone was enough to desorb the adsorbed neonicotinoid insecticides. Therefore, 2 × 5 mL of acetone was used for the remaining experiments.

Subsequently, different elution time at 2 min, 4 min, 6 min, 8 min, and 10 min were investigated. As shown in Figure 7g, the maximum peak areas of four neonicotinoid insecticides were achieved at 6 min. Therefore, elution time was set at 6 min in the following experiments.

**Figure 7 ijerph-20-00715-f007:**
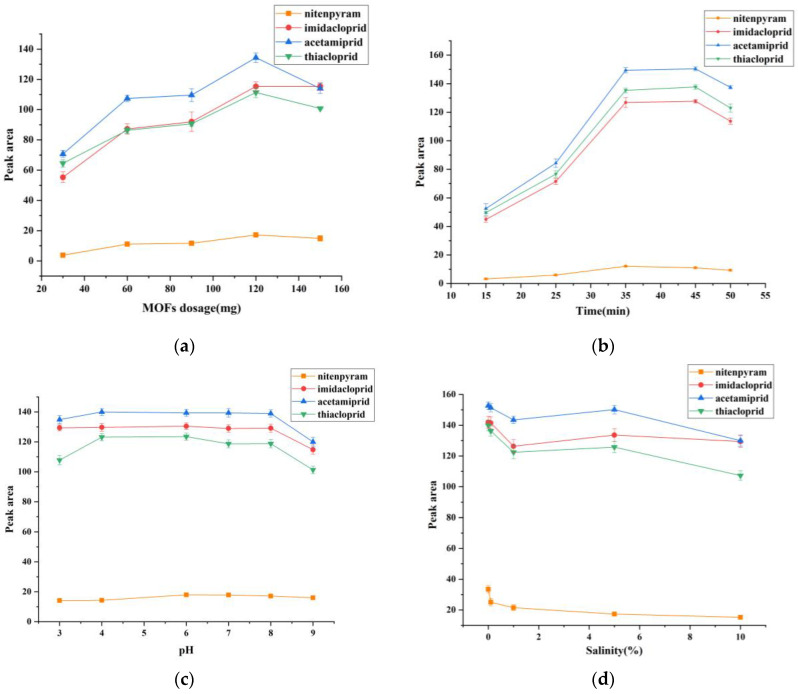
Effects of (**a**) dosage of MIL-53, (**b**) extraction time, (**c**) sample solution pH, (**d**) salinity, (**e**) eluent type, (**f**) eluent volume, and (**g**) elution time on the DME efficiencies for four neonicotinoid insecticides. Extraction conditions: sample volume, 50 mL; concentration of each neonicotinoid insecticide: 10 μg L^−1^. (**a**) Water sample pH: 6.0; water sample volume: 100 mL; salt concentration: 0%; extraction time: 25 min; eluent: 2 × 5 mL of acetone; elution time: 2 × 10 min; (**b**) Water sample pH: 6.0; water sample volume: 100 mL; salt concentration: 0%; MOF dosage: 120 mg; eluent: 2 × 5 mL of acetone; elution time: 2 × 10 min; (**c**) Water sample volume: 100 mL; salt concentration: 0%; MOF dosage: 120 mg; extraction time: 35 min; eluent: 2 × 5 mL of acetone; elution time: 2 × 10 min; (**d**) Water sample pH: 6.0; water sample volume: 100 mL; MOF dosage: 120 mg; extraction time: 35 min; eluent: 2 × 5 mL of acetone; elution time: 2 × 10 min; (**e**) Water sample pH: 6.0; water sample volume: 100 mL; salt concentration: 0%; MOF dosage: 120 mg; extraction time: 35 min; eluent volume: 2 × 5 mL; elution time: 2 × 10 min; (**f**) Water sample pH: 6.0; water sample volume: 100 mL; salt concentration: 0%; MOF dosage: 120 mg; extraction time: 35 min; eluent: acetone; elution time: 2 × 10 min; (**g**) Water sample pH: 6.0; water sample volume: 100 mL; salt concentration: 0%; MOF dosage: 120 mg; extraction time: 35 min; eluent: 2 × 5 mL of acetone.

### 3.4. Regeneration of MIL-53 MMM for DME

Reusability of the adsorbents related to the efficiency and cost effectiveness, which is crucial for its practical applications. The exhausted MIL-53 MMM was rinsed by acetone prior to the next DME process. As shown in Figure 8, the reusability efficiency of the MIL-53 MMM remained high after fourteen successive cycles. The results suggested that the MIL-53 MMM based DME pretreatment method possessed high stability and good reusability.

### 3.5. Analytical Performance of the DME-HPLC Method

Under the optimum extraction conditions, several analytical performance parameters of this established method were evaluated. As listed in Table 1, good linearity was achieved within the concentration ranges of 0.20–15.00 μg L^−1^ for nitenpyram, 0.04–15.00 μg L^−1^ for imidacloprid and thiacloprid, and 0.05–15.00 μg L^−1^ for acetamiprid. Calibration curves were obtained by plotting the peak areas (y) of neonicotinoid insecticides versus their concentrations (x). The obtained correlation coefficients (*r*^2^) is between 0.990–0.996. The limits of detection (LODs) and the limits of quantitation (LOQs) were calculated based on analyte signal to background noise ratio of 3 and 10, respectively. As seen in Table 1, the LODs and LOQs of four neonicotinoid insecticides were 0.013–0.064 μg L^−1^ and 0.038–0.190 μg L^−1^, respectively.

The relative standard deviations (RSDs) of peak areas at three spiked concentration levels (0.5, 5 and 10 μg L^−1^) were used to evaluate the precision of the proposed method. As listed in Table 2, the spiked recoveries ranged from 78.72% to 119.68%. The intra-day (*n* = 6) and inter-day (*n* = 6) RSDs were in the range of 3.07–12.78% and 3.43–13.12%, respectively.

### 3.6. Application of the DME-HPLC Method to Real Water Samples

Three real water samples collected from the laboratory tap water, Qingdao Bohai Bay seawater, Qingdao Yinfu Reservoir surface water were used to verify the practicability of this proposed method. Representative HPLC chromatograms of tap water samples were displayed in Figure 9. None of these four neonicotinoid insecticides were detected in above water samples. Three levels of spiked samples at 0.5, 5 and 10 μg L^−1^ were used to investigate the extraction recoveries of the developed method. As listed in Table 3, satisfactory recoveries of four neonicotinoid insecticides were achieved in the range of 72.50–117.98%. These results demonstrated that the proposed method can be successfully applied to the enrichment and determination of neonicotinoid pesticides in environmental water samples.

### 3.7. Method Performance Comparison

The analytical performance of the developed DME-HPLC method was compared with other reported methods for the analysis of neonicotinoid insecticides. As shown in Table 4, compared with various pretreatment methods combining HPLC-DAD as the detector, the MOFs MMM-based DME method exhibits lower LODs and LOQs, on the other hand, in comparison with those pretreatment methods using MOFs as adsorbents, the developed MIL-53 MMM offers similar sensitivity with UiO-66 dispersive solid phase extraction followed by HPLC-MS/MS method. In addition, the MIL-53 MMM possesses the highest reusability and convenient separation process, suggesting the high practicability of the MIL-53 MMM based DME method.

## 4. Conclusions

In the present work, MIL-53 MMM was successfully prepared and applied for DME of four neonicotinoid insecticides in environmental water samples along with HPLC-DAD determination. The specific structure of MIL-53 offered abundant adsorption sites leading to high extraction efficiencies and the membrane form of MOFs simplified the separation procedure. The developed MOFs-DME pretreatment combined with the HPLD-DAD method exhibited high sensitivity with low LODs and LOQs. Meanwhile, satisfactory precision and accuracy were achieved by this established DME-HPLC method, indicating that the developed MIL-53 MMM is a highly efficient and reliable material for the enrichment of neonicotinoid insecticides in environmental waters.

## Figures and Tables

**Figure 1 ijerph-20-00715-f001:**
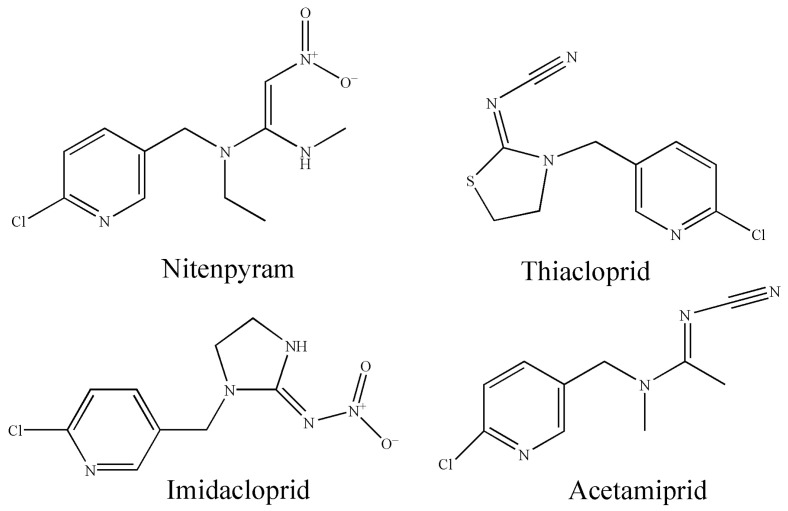
Chemical structures of four neonicotinoid insecticides.

**Figure 2 ijerph-20-00715-f002:**
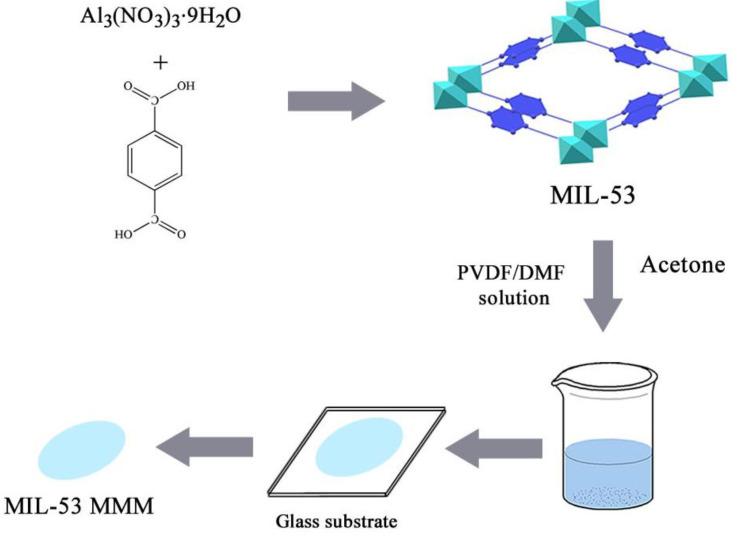
Illustration for the synthesis of the MIL-53 MMM.

**Figure 3 ijerph-20-00715-f003:**
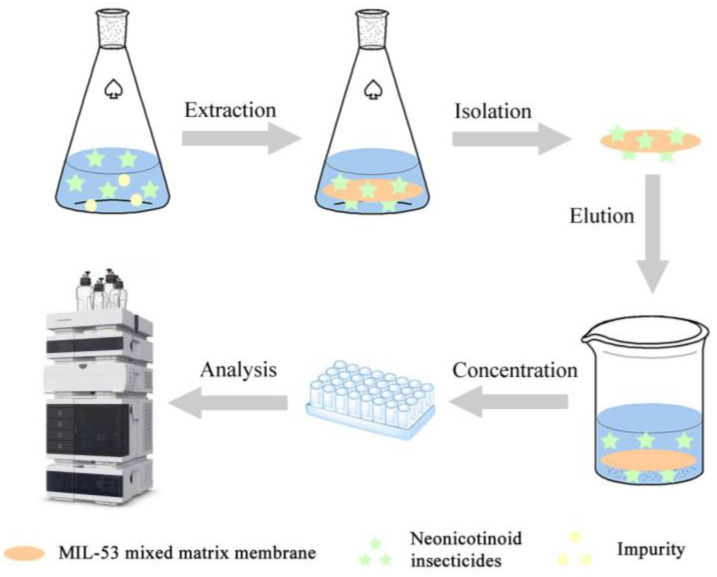
Schematic of the MIL-53 MMM based DME process.

**Figure 4 ijerph-20-00715-f004:**
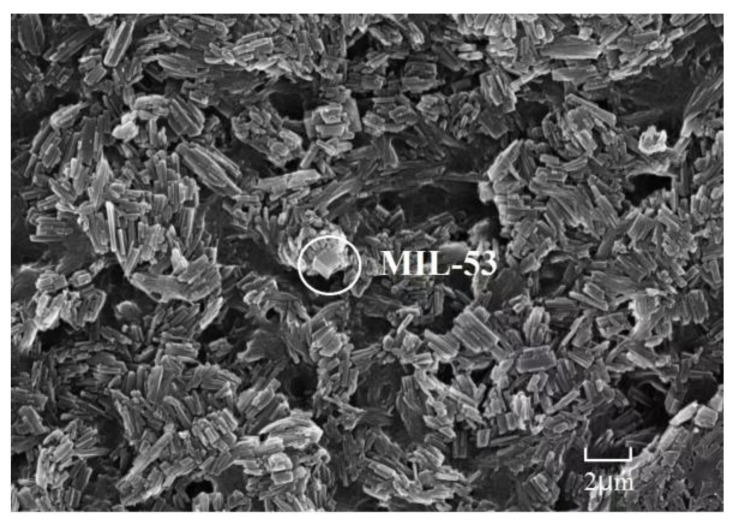
SEM images of MIL-53 MMM with the plotting scale of 2 μm.

**Figure 5 ijerph-20-00715-f005:**
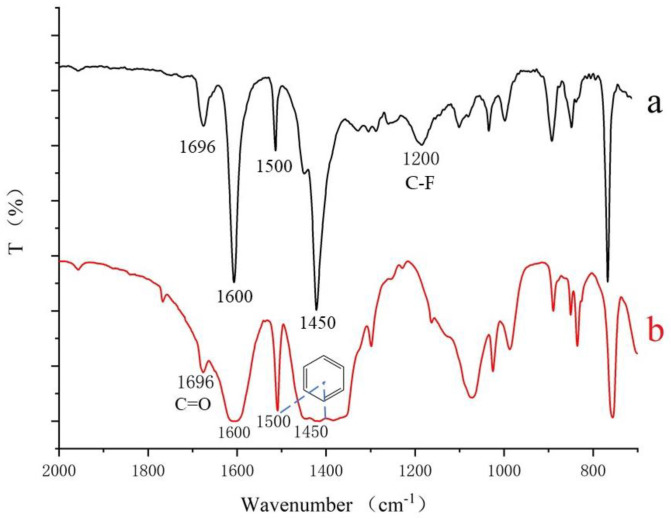
FTIR spectra of MIL-53 MMM (**a**), MIL-53 crystal (**b**).

**Figure 6 ijerph-20-00715-f006:**
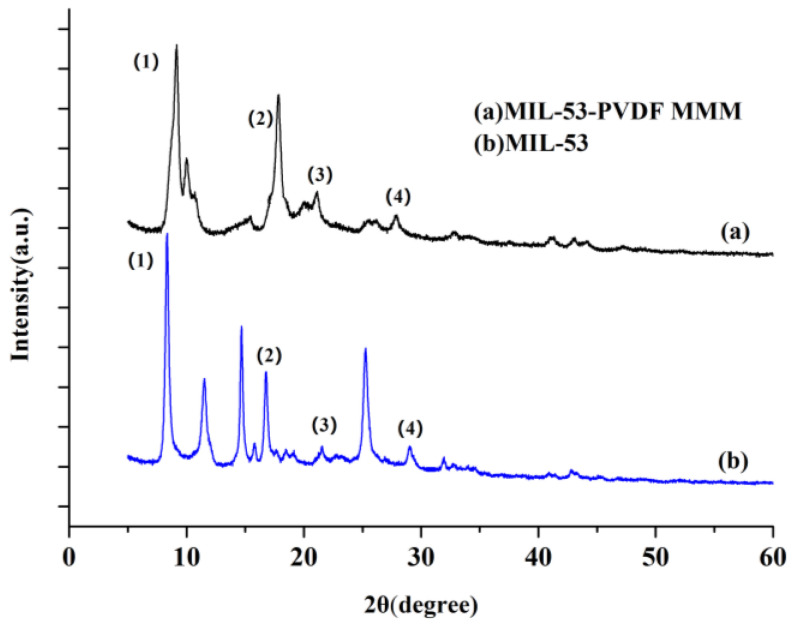
XRD patterns of MIL-53 MMM (**a**), MIL-53 crystal (**b**).

**Figure 8 ijerph-20-00715-f008:**
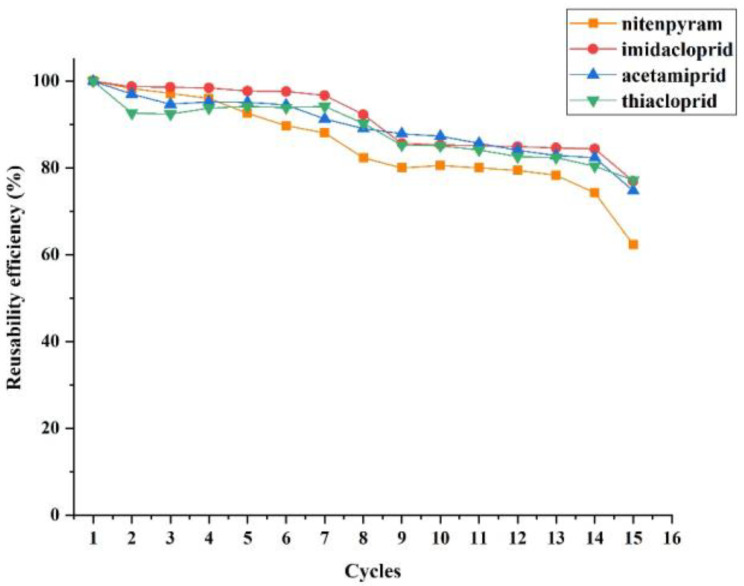
Reusability of the MIL-53 MMM for DME.

**Figure 9 ijerph-20-00715-f009:**
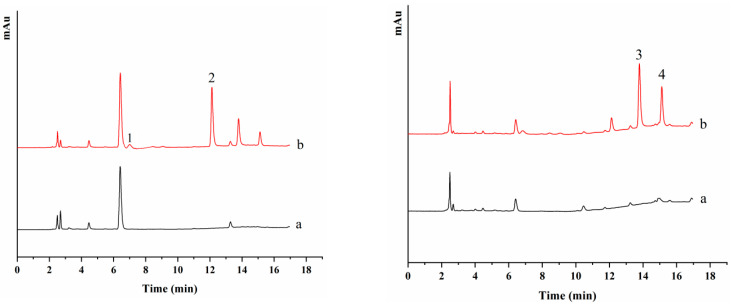
HPLC chromatograms from DAD of four neonicotinoid insecticides in real tap water samples after DME without spiking (a) and with spiking (b); measurement wavelength 270 nm (**left**) and measurement wavelength 244 nm (**right**). Peak identification: (1) nitenpyram (2) imidacloprid (3) acetamiprid and (4) thiacloprid.

**Table 1 ijerph-20-00715-t001:** Regression equation, linear range, correlation coefficient (*r*^2^) and method detection limit, and limit of quantification of four neonicotinoid insecticides.

Analyte	Regression Equation ^a^	Correlation Coefficient (*r*^2^)	Linear Range (µg L^−1^)	LOD (µg L^−1^)	LOQ (µg L^−1^)
Nitenpyram	y = 1.764x − 0.6893	0.9937	0.20–15.00	0.064	0.190
Imidacloprid	y = 12.801x − 1.0903	0.9957	0.04–15.00	0.013	0.038
Acetamiprid	y = 14.542x + 0.0519	0.9960	0.05–15.00	0.017	0.050
Thiacloprid	y = 12.638x − 2.8553	0.9901	0.04–15.00	0.014	0.041

^a^ x means concentration of neonicotinoid insecticides (µg L^−1^), y means peak area.

**Table 2 ijerph-20-00715-t002:** Intra-day and inter-day precision (RSDs, %) for the MOF membrane extraction-HPLC method for the determination of four neonicotinoid insecticides.

Insecticides	Spiked (µg L^−1^)	Intra-Day (*n* = 6)		Inter-Day (*n* = 6)	
		Recovery (%)	RSD (%)	Recovery (%)	RSD (%)
Nitenpyram	0.50	112.17	12.04	111.03	12.55
	5.00	104.19	10.04	87.97	9.02
	10.00	84.52	5.06	81.12	4.44
Imidacloprid	0.50	111.25	3.07	119.68	13.12
	5.00	104.27	3.95	105.48	6.37
	10.00	96.84	3.72	90.88	3.43
Acetamiprid	0.50	92.53	8.53	80.43	12.65
	5.00	85.02	4.19	90.08	4.92
	10.00	92.37	3.65	80.83	7.04
Thiacloprid	0.50	106.11	12.78	96.14	11.97
	5.00	95.40	6.97	88.28	6.39
	10.00	92.86	3.62	78.72	9.82

**Table 3 ijerph-20-00715-t003:** Determination of four neonicotinoid insecticides and method recoveries in real water samples.

Insecticides	Spiked (μg L^−1^)	Tap Water		Surface Water		Sea Water	
		Found (μg L^−1^)	Recovery (% ± RSD, *n* = 3)	Found (μg L^−1^)	Recovery (% ± RSD, *n* = 3)	Found (μg L^−1^)	Recovery (% ± RSD, *n* = 3)
Nitenpyram	0.00	ND		ND		ND	
	0.50	0.43	86.09 ± 12.91	0.57	114.06 ± 11.96	0.58	115.57 ± 10.93
	5.00	3.96	79.24 ± 5.10	5.34	106.83 ± 6.31	5.54	110.99 ± 6.12
	10.00	12.1	72.50 ± 11.58	9.03	90.26 ± 4.93	8.80	88.00 ± 13.25
Imidacloprid	0.00	ND		ND		ND	
	0.50	0.40	79.53 ± 12.66	0.56	112.33 ± 7.9	0.57	114.94 ± 8.03
	5.00	3.88	77.64 ± 3.59	4.41	88.10 ± 3.83	5.47	109.45 ± 2.28
	10.00	7.38	73.81 ± 10.59	7.51	75.14 ± 5.79	10.29	102.88 ± 3.46
Acetamiprid	0.00	ND		ND		ND	
	0.50	0.51	101.06 ± 10.31	0.45	90.52 ± 11.11	0.38	75.85 ± 7.26
	5.00	3.96	79.14 ± 3.30	4.01	80.25 ± 7.36	5.69	113.85 ± 3.33
	10.00	7.42	74.23 ± 4.62	8.11	81.11 ± 3.59	9.50	94.95 ± 8.70
Thiacloprid	0.00	ND		ND		ND	
	0.50	0.59	117.98 ± 13.76	0.59	117.98 ± 7.16	0.57	113.76 ± 9.33
	5.00	3.76	75.26 ± 5.98	4.20	83.96 ± 5.77	5.79	115.82 ± 4.62
	10.00	7.66	76.56 ± 1.08	8.22	82.18 ± 5.64	9.90	98.95 ± 4.13

**Table 4 ijerph-20-00715-t004:** Method performance comparison for neonicotinoid insecticides by HPLC in water samples.

Insecticides	Pretreatment Technique	Adsorbents	Detection Techniques	LOD (μg L^−1^)	LOQ (μg L^−1^)	Reusability	Refs.
Imidacloprid, acetamiprid, thiacloprid, thiamethoxam	Magnetic solid phase extraction	Magnetic nanoporous carbon	HPLC-UV	0.01–0.06	NA	NA	[47]
Imidacloprid, acetamiprid, thiamethoxam thiacloprid	Magnetic solid phase extraction	Magnetic porous carbon	HPLC-UV	0.1–0.2	NA	NA	[48]
Dinotefuran, thiamethoxam, clothianidin, imidacloprid, acetamiprid, thiacloprid	Magnetic solid phase extraction	Magnetic zeolitic imidazolate framework/grapheme oxide	HPLC-MS/MS	0.06–1.0	0.2–3.0	NA	[15]
Dinotefuran, thiamethoxam, clothianidin, imidacloprid, acetamiprid and thiacloprid	Dispersive solid phase extraction	UiO-66	HPLC-MS/MS	0.02–0.4	0.05–1	NA	[14]
Idacloprid, acetamiprid, thiacloprid, nitenpyram	Dispersion membrane extraction	MIL-53(Al) MMM	HPLC-DAD	0.01–0.06	0.038–0.19	13	This work

NA means not accessible.

## Data Availability

All data and materials are available in the manuscript.

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
