# Peer review of "Determination of Neonicotinoid Insecticides in Environmental Water by the Enrichment of MIL-53 Mixed Matrix Membrane Coupled with High Performance Liquid Chromatography"

_ijerph, 2022, doi:10.3390/ijerph20010715_

Round 1
Reviewer 1 Report
The authors explored a new material mixed matrix membrane (MOF-MMMs), which were synthesized and applied for dispersive membrane extraction (DME) of four neonicotinoid insecticides. Furthermore, several experimental conditions were optimized in detail, involving dosage percentage of MOF, extraction time, sample pH, salinity, type and volume of eluent and elution time. This work is nice and will get more interestingly for the reader, but the below
1. Revise the figure caption
2. In fig4, the main functional groups should be labeled in the map.
3. Please provide the PXRD of MIL-53 after the extraction (DME) of four neonicotinoid insecticides
4. Metal-organic frameworks (MOFs) are a novel class of porous materials composed of organic ligands with inorganic clusters. Due to their unique properties, such as high specific surface area, tunable pore size and easy functionalization, MOFs have attracted extensive attention and have been widely applied in various fields such as gas storage, catalysis, sensing and especially in sample preparation. There should highlight and cite updated documents, such as J. Alloy. Compd, 2022, 897, 163178; Inorganics, 10(2022) 202; Micropor. Mesopor. Mat, 341(2022) 112098; and Mater. Today. Commum., 2022, 31,103514.
5. The author should check the recovery rate and accuracy of the sample.
6. The structural stability should be confirmed by the morphology of MIL-based samples after multiple adsorption tests.
7. From the design of the samples, the authors have to compare the high sensitivity with limits of detection and quantification between this material and similar work, in which will highlight your work.
Author Response
Detailed Reponses to Comments on ijerph-2082410
We highly appreciate the Reviewers’ valuable comments, significant guidance and sincere contributions. The manuscript has been carefully thoroughly revised in accordance with the comments. We have fully addressed the concerns of the reviewers and the main revisions have been given, which are marked up by “Track Changes” mode in the revised manuscript. Additionally, the total manuscript has been carefully checked, altered and polished for improvement.
Response to Reviewer 1 Comments:
Reviewer 1: Comments and Suggestions for Authors
The authors explored a new material mixed matrix membrane (MOF-MMMs), which were synthesized and applied for dispersive membrane extraction (DME) of four neonicotinoid insecticides. Furthermore, several experimental conditions were optimized in detail, involving dosage percentage of MOF, extraction time, sample pH, salinity, type and volume of eluent and elution time. This work is nice and will get more interestingly for the reader, but the below
Point 1: Revise the figure caption.
Response 1: Thanks very much for the kind reminder. We revised the caption of Figure 7. The majuscule captions A to G were revised to lowercase, which is consistent with the images. Related descriptions are now revised in the manuscript for clarity (Page 10, line 267-281). Besides, the figures previous in supplementary materials were shown in the revised manuscript.(Page 4, line 153-154, updated Figure. 2; Page 10, line 289-190, updated Figure. 8; Page 12-13, line 327-332, updated Figure 9)
Point 2: In fig4, the main functional groups should be labeled in the map.
Response 2: Thanks very much for the kind suggestion. The main functional groups including benzene ring, C=O, C-F which are related to the adsorption peaks were supplemented in Figure 5 (previous Figure 4). (Page 7, line 196, updated Figure. 5)
Point 3: Please provide the PXRD of MIL-53 after the extraction (DME) of four neonicotinoid insecticides
Response 3: Thanks very much for the kind suggestion. We are so sorry that we cannot provide the PXRD of MIL-53 after the extraction, we did not measure the PXRD of the exhausted MIL-53 when investigating the reusability of the adsorbents, it is time-deficient to supplement this characterisation within 5 days, under the COVID condition in China. It can be deduced that the MIL-53-MMM possesses high stability, due to its excellent reusability. The adsorption mechanism of MIL-53-MMM has been proven to be a combination of π-π conjugation, cationic-π bonding and flexible skeleton, if the MIL-53-MMM were to collapse, it could not be reused 14 times. Therefore, the excellent reusability suggested the good stability of MIL-53-MMM.
Point 4: Metal-organic frameworks (MOFs) are a novel class of porous materials composed of organic ligands with inorganic clusters. Due to their unique properties, such as high specific surface area, tunable pore size and easy functionalization, MOFs have attracted extensive attention and have been widely applied in various fields such as gas storage, catalysis, sensing and especially in sample preparation. There should highlight and cite updated documents, such as J. Alloy. Compd, 2022, 897, 163178; Inorganics, 10(2022) 202; Micropor. Mesopor. Mat, 341(2022) 112098; and Mater. Today. Commum., 2022, 31,103514.
Response 4: Thanks very much for the kind suggestion. The valueble references mentioned above were supplemented as [18], [19], [22], [26] in the revised manuscript. (Page 2, line 63-64)
Point 5: The author should check the recovery rate and accuracy of the sample.
Response 1: Thanks very much for the kind reminder. The spiked recoveries ranges were investigated and listed in Table 2. Besides, in section 3.6, recoveries of four neonicotinoid insecticides in real water sampleswere were achieved in the range of 72.50–117.98%. Related descriptions are now added in the revised manuscript for clarity, as follows,
“As listed in Table 2, the spiked recoveries ranges from 78.72% to 119.68%.” (Page 11, line 308)
Point 6: The structural stability should be confirmed by the morphology of MIL-based samples after multiple adsorption tests.
Response 6: Thanks very much for the kind suggestion. As explained in Response 3, we are so sorry that we cannot provide the morphology of MIL-53 after the extraction, we did not measure the morphology of the exhausted MIL-53 when investigating the reusability of the adsorbents, it is time-deficient to supplement this characterisation within 5 days, under the COVID condition in China. It can be deduced that the MIL-53-MMM possesses high stability, due to its excellent reusability. The adsorption mechanism of MIL-53-MMM has been proven to be a combination of π-π conjugation, cationic-π bonding and flexible skeleton, if the MIL-53-MMM were to collapse, it could not be reused 14 times. Therefore, the excellent reusability suggested the good stability of MIL-53-MMM.
Point 7: From the design of the samples, the authors have to compare the high sensitivity with limits of detection and quantification between this material and similar work, in which will highlight your work.
Response 7: Thanks very much for the kind suggestion. The sensitivity of this method was compared with other reported methods in Section 3.7 and Table 4. As suggested, the limits of quantification were supplemented in Table 4. (Page 13, line 350)
As shown in Table 4, compared with various pretreatment methods combining HPLC-DAD as the detector, the MOFs MMM based DME method exhibits lower limits of detection, on the other hand, in comparison with those pretreatment methods using MOFs as adsorbents, the developed MIL-53 MMM offers similar sensitivity with UiO-66 dispersive solid phase extraction followed by HPLC-MS/MS method. These results suggest the high sensitivity of our method.

Reviewer 2 Report
This manuscript synthesized MIL-53(Al) mixed matrix membrane for dispersive membrane extraction of four neonicotinoid insecticides. The experimental data is detailed, and the manuscript was well-organized. In order for this work to be assumed as a helpful for the research community, the following issues could be added to the discussion.
1. Section 2.4:
(1) The presentation of elution volume is 5´2 mL, however, in the following sections, it is presented as 2´5 mL, please confirm the number of elution process.
(2) The elution time could be displayed in this section.
2. Table 1: The correlation coefficient r2 should be a uniform layout.
3. Table 3: The abbreviation of these analytes in the caption of this table needs to be revised.
4. In the section of introduction, the advances should be improved. eg: Chinese Journal of Catalysis, 2022, 43, 2652–2664, etc.
5. Section 3.2: the plotting scale in Figure 3 is not clear.
6. Why was PVDF selected as the polymer to construct the MOFs mixed matrix membrane?
Author Response
Detailed Reponses to Comments on ijerph-2082410
We highly appreciate the Reviewers’ valuable comments, significant guidance and sincere contributions. The manuscript has been carefully thoroughly revised in accordance with the comments. We have fully addressed the concerns of the reviewers and the main revisions have been given, which are marked up by “Track Changes” mode in the revised manuscript. Additionally, the total manuscript has been carefully checked, altered and polished for improvement.
Response to Reviewer 2 Comments:
Reviewer 2: Comments and Suggestions for Authors
This manuscript synthesized MIL-53(Al) mixed matrix membrane for dispersive membrane extraction of four neonicotinoid insecticides. The experimental data is detailed, and the manuscript was well-organized. In order for this work to be assumed as a helpful for the research community, the following issues could be added to the discussion.
Point 1: Section 2.4:
(1) The presentation of elution volume is 5´2 mL, however, in the following sections, it is presented as 2´5 mL, please confirm the number of elution process.
(2) The elution time could be displayed in this section.
Response 1 : Thanks very much for the kind suggestion. As suggested, elution volume was revised as 2´5 mL in Section 2.4, which means the elution process was conducted twice.
The elution time was supplemented in this section. Related descriptions are now added in the revised manuscript for clarity, as follows,
“After extraction, 2´5 mL acetone was used to elute the adsorbed insecticides from MIL-53 MMM, elution time was 6 min” (Page 4, line 159-160)
Point 2: Table 1: The correlation coefficient r2 should be a uniform layout.
Response 2: Thanks very much for the kind reminder. The correlation coefficient in the caption of Table 1 was revised to lowercase r2. (Page 11, line 303)
Point 3: Table 3: The abbreviation of these analytes in the caption of this table needs to be revised.
Response 3: Thanks very much for the kind reminder. The name of these analytes were revised to four neonicotinoid insecticides in the caption of Table 3. (Page 12, line 325)
Point 4: In the section of introduction, the advances should be improved. eg: Chinese Journal of Catalysis, 2022, 43, 2652–2664, etc.
Response 4: Thanks very much for the kind suggestion. The valueble references mentioned above were supplemented as [32] in the revised manuscript. (Page 2, line 74)
Point 5: Section 3.2: the plotting scale in Figure 3 is not clear.
Response 5: Thanks very much for the kind suggestion. The plotting scale in Figure 4 (previous Figure 3) is revised to a clearer one. (Page 6, line 185, updated Figure. 4)
Point 6: Why was PVDF selected as the polymer to construct the MOFs mixed matrix membrane?
Response 6: Thanks very much for the kind suggestion. We consulted several reported studies [Ref 1-3], PVDF has been proven to be an excellent polymer martix to construct MOFs mixed matrix membrane. It exhibited good practicability in decontamination and gas separation, due to its high stability and mechanical strength. Thus, PVDF was selected as the polymer to construct the MIL-53 mixed matrix membrane.
Reference
[Ref. 1] Denny, M. S., Cohen, S. M., In Situ Modification of Metal–Organic Frameworks in Mixed-Matrix Membranes. Angewandte Chemie International Edition, 2015, 54, 9029–9032.
[Ref. 2] H. Rajati, A.H. Navarchian, D. Rodrigue, S. Tangestaninejad, Improved CO2 transport properties of Matrimid membranes by adding amine-functionalized PVDF and MIL-101(Cr), Separation and Purification Technology, 2020, 235, 116149.
[Ref. 3] H. Rajati, A.H. Navarchian, S. Tangestaninejad, Preparation and characterization of mixed matrix membranes based on Matrimid/PVDF blend and MIL-101(Cr) as filler for CO2/CH4 separation, Chemical Engineering Science, 2018, 185, 92-104.
